# Comprehensive Comparison of the Capacity of Functionalized Sepharose, Magnetic Core, and Polystyrene Nanoparticles to Immuno-Precipitate Procalcitonin from Human Material for the Subsequent Quantification by LC-MS/MS

**DOI:** 10.3390/ijms241310963

**Published:** 2023-06-30

**Authors:** Thomas Masetto, Kai Matzenbach, Thomas Reuschel, Sebastian-Alexander Tölke, Klaus Schneider, Lea Marie Esser, Marco Reinhart, Laura Bindila, Christoph Peter, Matthias Grimmler

**Affiliations:** 1Institute of Molecular Medicine I, Medical Faculty, Heinrich Heine University Düsseldorf, 40225 Düsseldorf, Germany; thmas100@uni-duesseldorf.de (T.M.);; 2DiaSys Diagnostic Systems GmbH, Alte Straße 9, 65558 Holzheim, Germany; 3Institute for Biomolecular Research, Hochschule Fresenius gGmbH, University of Applied Sciences, Limburger Straße 2, 65510 Idstein, Germany; 4Clinical Lipidomics Unit, Institute of Physiological Chemistry, University Medical Center, 55131 Mainz, Germany; 5GfA GmbH, Allgäuer Straße 1, 87459 Pfronten, Germany; 6DiaServe Laboratories GmbH, Seeshaupter Straße 27, 82393 Iffeldorf, Germany

**Keywords:** sepsis, procalcitonin, LC-MS/MS quantification, standardization, immuno-enrichment, biopolymers, PETIA, magnetic particles, Sepharose particles, polystyrene

## Abstract

Sepsis is a life-threatening organ dysfunction caused by a dysregulated host response to infection. The fast and accurate diagnosis of sepsis by procalcitonin (PCT) has emerged as an essential tool in clinical medicine. Although in use in the clinical laboratory for a long time, PCT quantification has not yet been standardized. The International Federation of Clinical Chemistry working group on the standardization of PCT (IFCC-WG PCT) aims to provide an LC-MS/MS-based reference method as well as the highest metrological order reference material to address this diagnostic need. Here, we present the systematic evaluation of the efficiency of an immuno-enrichment method, based on functionalized Sepharose, magnetic-core, or polystyrene (latex) nano-particles, to quantitatively precipitate PCT from different human sample materials. This method may be utilized for both mass spectrometric and proteomic purposes. In summary, only magnetic-core nano-particles functionalized by polyclonal PCT antibodies can fulfil the necessary requirements of the international standardization of PCT. An optimized method proved significant benefits in quantitative and specific precipitation as well as in the subsequent LC-MS/MS detection of PCT in human serum samples or HeLa cell extract. Based on this finding, further attempts of the PCT standardization process will utilize a magnetic core-derived immuno-enrichment step, combined with subsequent quantitative LC-MS/MS detection.

## 1. Introduction

Sepsis was responsible for almost 20% of all global deaths in 2017 [1]. Approximately one-third of septic patients will also die in consequence [1]. For this, fast and accurate diagnosis is necessary to initiate correct treatment [2,3,4,5]. Every hour of delayed diagnosis will raise sepsis-related mortality by 7% [6,7]. Besides bacterial or viral-caused inflammation, the occurrence of antibiotic-multi-resistant bacteria, as well as the age shift of the human population, will further increase the proportion of sepsis-derived mortality in the future. Consequently, the management of sepsis, its diagnosis, care, and monitoring are considered to be one of mankind’s main goals by the WHO and the UN [8].

The 12.5 kDa protein procalcitonin (PCT) has emerged as a powerful diagnostic tool in laboratory medicine to help define and differentiate a bacterial background in systemic inflammation [9,10,11]. PCT is present in healthy individuals in very low concentrations of less than 0.1 ng/mL and can rise by factor 1000 during the progression of a bacterial infection [9,12,13]. Besides its diagnostic utility [14,15,16,17], PCT is also used as a prognostic tool, providing powerful help in monitoring the success and duration of antibiotic medication [14,18,19,20].

To date, a broad variety of different chemiluminescence (CLIA) and particle-enhanced turbidimetric (PETIA)-based immunoassays to measure PCT exist on the market [21]. Besides the methodical differences between CLIA and PETIA, the single or combined use of monoclonal or polyclonal antibodies also differs significantly between these assays [21,22]. So far, neither a reference material to standardize and trace PCT measurements, nor a higher-order reference method exist to quantify the absolute amount of PCT in human samples. The current heterogeneity of PCT measurements in clinical diagnostics is also reflected by the regular monitoring of national external quality assessment programs (EQAs) [23,24].

Intensive work to define, harmonize, and standardize PCT quantification is currently organized by the IFCC-WG PCT [21,23,25,26]. Part of this activity is the identification and evaluation of methods to specifically enrich and precipitate PCT from human samples and the subsequent quantification by LC-MS/MS of the total PCT content of the sample. Beside solid-phase extraction methods [25,26], the evaluation of immuno-enrichment techniques is also part of the IFCC-WG PCT program [27].

A robust quantification and reference method should be able to detect the respective analyte with sufficiently high precision even in the very low concentration range of 0.1–0.05 ng/mL. Despite the enormous progress in the field of direct quantitative detection using modern mass-spectrometric methods, it is not yet possible to quantify such a low abundant protein concentration without upstream enrichment steps. Various techniques are currently used for appropriate protein enrichment in mass spectrometry. All of them must meet specific requirements for absolute metrological quantification and must be robust enough to enable reproducible and transferable results in different laboratories. Furthermore, they need to be applicable to cover the entire medically relevant concentration range of an analyte [28]. Additionally, they must be capable of quantitatively capturing the analyte within a suitable amount of time (1–4 h) and also come along with acceptable effort and material costs, without reducing the sensitivity and precision of the downstream analytical mass spectrometry method. In particular, proteins, such as PCT with various isoforms [12,29,30], putative modulations of the analyte by age, gender, ethnicity, state of health, or other post-translational modifications as well as protein degradation, represent an enormous challenge for absolute quantification in medical diagnostics. A solid upfront characterization of the used components, especially of antibodies and particles but also of applied processes, is the basis of a sensitive and reproducible method of absolute PCT quantification.

In this work, we present the first comprehensive evaluation of three immuno-functionalized nano-material carrier systems (Sepharose, magnetic core, polystyrene/latex nano-particles) and their suitability for the robust precipitation and quantitative depletion of PCT from human serum or cell extract. PCT-specific polyclonal antibodies were evaluated and quantified for immuno-functionalization of the respective nano-particles. Accordingly, in addition to quantitative enrichment, the specific detection of the analyte with minimal bias by non-specific background was assessed. The focus of the presented analysis is the homogeneity of the particles regarding a reproducible covalent coupling of the PCT-specific antibody, the robust and quantitative depletion of the analyte from the sample, the loss-free enrichment of the immobilized PCT antigen, as well as the suitability of the respective nano-particles for a maximal sensitive quantification of the subsequently tryptic eluted peptides by LC-MS/MS technology.

## 2. Results

### 2.1. Covalent Coupling of Antibodies to Nano-Particles

The three nano-particles characterized in this work (PGS, MagP, and Lx) employ different binding schemes to capture the antibodies. In Table 1, the binding as well as the main characteristics of the three different nano-particles are summarized.

On the one hand, PGS and MagP bind the antibody through its Fc fragment, offering a better orientation (F(ab)2 up) to recognize the antigen. On the other hand, Lx particles covalently couple antibodies by a reaction with the antigen binding site of the F(ab)2 [31]. Potentially, this could lead to a lower binding capacity, as one of the two binding sites of each antibody is sterically impeded. This should in case be compensated by higher amount of functionalized Lx. The typical working set of the used particles is shown in Appendix A. Respective amounts of particles were chosen according to their individual antibody-binding capacity to enable fast and complete precipitation in an acceptable time range (1–4 h) and by the use of universally available lab devices (e.g., tabletop centrifuge). The particle to antibody coupling and the immuno-precipitation experiments were optimized and repeated (min. *n* = 3) to ensure robustness of processes and reported results.

### 2.2. Nano-Particle Characterization: Structure and Size Distribution Evaluation

For routine handling in metrological quantification, the workflow needs to be simple, robust, and normalized. The homogeneity of used material is a prerequisite to ensure loss-free handling during precipitation and further steps (washing, tryptic digest, and elution). Figure 1 shows the visual characterization of the three nano-particles using an optical microscope. In the case of the Lx particle, the diameter range is too small to be reliably determined with the software of the microscope. Its dimensions and size distribution (0.350 µm, ±0.0072 µm) were measured through Dynamic Light Scattering (DLS). By overall small size, Lx particles have a very large surface that may be immuno-functionalized for very fast binding and depletion of an antigen.

A second significant observation is the porosity of the PGS particles, evident upon incubation with Coomassie brilliant blue dye, whereas the other two nano-materials are not colored (Figure 1, compare c to d and e). Besides the higher specific binding capacity of linked antibodies, the porosity of Sepharose material may in turn lead to higher retention time in washing steps and an increase in the unspecific binding/capture of undesired components from the sample in the subsequent analysis. Moreover, strong heterogeneity of particle size is evident by Sepharose material (Figure 1a,c), hindering optimal and comparable binding kinetics and the application of optimal, loss-free centrifugation forces during the processing of the sample.

In clear contrast, MagP´s diameter distribution appears to be rather regular (almost monodispersed), comparable to that of the Lx particles (Figure 1, compare b, d to e). In summary, MagP and Lx particles seem to provide suitable characteristics for strongly normalized workflow processes, whereas Sepharose in this characterization appeared to be less homogeneous (refer to Figure 1).

### 2.3. Calculation of Binding Capacities, Repeatability, and Characterization of Depletion Kinetics

For metrological quantification, the depletion of the antigen from a biological sample needs to be as complete as possible. The establishment of a suitable and robust immunodepletion system requires a comprehensive selection of components and the optimization of used integral parts and workflow in advance. Moreover, the separation in distinct steps of the manufacturing depletion system itself and of the subsequent precipitating activity should be made possible to ensure suitable handling in a routine laboratory within the regular working process.

In the initial step, the amount of polyclonal antibody, the volume of dispersed nano-particles, the covalent cross-link, and the blocking process for the three kinds of particles were varied and optimized. The analysis of depletion efficiency of PCT-specific antibodies was initially performed by SDS-PAGE gel and Coomassie staining, using highly purified rhPCT, spiked into human serum (Figure 2). The amount of antibody to be used, a covalent vs. a non-covalent antibody crosslink, as well as the amount of antigen/saturation of the respective precipitating system, were optimized in subsequent steps (compare Figure 2a to Figure 2b,c to Figure 2d).

In the case of PGS, whose antibody binding capacity is very high (approx. 60 times higher than MagP, see Table 1), the optimization focused on the highest amount of rhPCT possible to precipitate with the same number of particles, both non-covalently and covalently bound (Figure 2a,b). For this reason, the volume of PGS was kept constant throughout the experiments (15 µL), reaching a maximal recovery of approx. 2 µg rhPCT (Figure 2a,b).

On the other hand, due to their lower antibody binding capacity, the optimization for the MagP involved an increasing number of particles (volume varying between 10 and 50 µL) to precipitate the same input quantity of rhPCT (Figure 2c,d). The antigen recovery of 50 µL MagP was approx. 1 µg.

Comparably, the Lx optimization also involved increasing particle quantities (volume varying between 100 and 200 µL), and in this case the limiting step is the antigen binding capacity, as previously mentioned. Similarly to the MagP, the antigen recovery of the Lx was also approx. 1 µg with working volumes between 150 and 200 µL.

In conclusion, from this preliminary semi-quantitative evaluation (Figure 2), all the particles can precipitate rhPCT antigen amounts of at least 1 µg. This can be achieved with a non-covalent as well as a covalent coupling strategy of anti-PCT antibodies. The optimized immuno-precipitation amounts for the three particles are 15 µL for PGS, 50 µL for MagP, and 150 µL for Lx. The exact immuno-precipitation capacities were analyzed by iBright instruments using 0.5 µg of rhPCT (refer to Figure 3).

The reproducibility and overall pull-down efficiency of all immuno-functionalized nano-particles were assessed by SDS-PAGE and Coomassie staining of the immunoprecipitated rhPCT, in reference to a normalized quantity of rhPCT (0.5 µg) on the same gel using iBright software (Figure 3 and Appendix A). The ability to precipitate and recover rhPCT from spiked human serum solutions (1 mL) was carried out in completely independent duplicates for each nano-particle (Figure 3, compare lanes 4 and 5, 6 and 7, 8 and 9). The quantitation was performed in duplicate to allow all the immuno-precipitation experiments to be analyzed in the same SDS-PAGE gel, to reach higher reproducibility.

It is noteworthy that the recovery for the PGS (approx. 50%, see Appendix A) and Lx (approx. 80%, see Appendix A) was significantly lower (significance criteria 5%) than for the MagP. The precipitation behavior of both PGS and Lx particles may indicate a less efficient recovery and higher experiment-to-experiment variation, even though their amounts have been previously optimized. This can be explained through the difficulties in handling and working with these materials (see Appendix A), as they do not precipitate homogeneously and do not form a compact pellet upon centrifugation. Moreover, Lx particles, due to their small size (0.35 µm), do not efficiently precipitate upon the maximal centrifugation force (13,500 rpm) of a table-top centrifuge within 5 min of separation. Consequently, PGS and Lx particles can in part remain in suspension after centrifuging or can be resuspended by the centrifuge breaking during the repeated washing steps (3×). On the other hand, the MagP show excellent recovery and a very good overall agreement between the two repetitions. This confirms the ease of handling (through magnetic force) and the great precipitation efficiency of this kind of functionalized nanoparticle. The recovered rhPCT amount being higher than the input could be ascribable to imprecisions in the pipetting process. However, the conclusions resulting from this test undoubtedly indicate an easier and more robust handling for the MagP than for the PGS and Lx.

PGS, MagP, and Lx depletion efficiency was also quantitatively evaluated by immunoassay in the medical analytical range (0 to approx. 50 ng/mL), where SDS-PAGE resolution is not enough to allow the direct visualization of such low abundant protein amounts. The remaining rhPCT concentrations in the supernatant upon incubation at 2, 4, and 6 h were quantified by the PCT FS PETIA by DiaSys Diagnostic GmbH (Figure 4 and Appendix A). Due to the large surface of nano-particles, depletion rate and efficiency are almost 100% already upon 2 h of incubation (see percentage depletion efficiency in Appendix A). Lx and MagP show a similar fast and constant reduction of rhPCT that also results in highly comparable depletion kinetics (compare Figure 4b,c).

On the other hand, obtained PCT concentration strongly varies in the case of functionalized PGS particles. As previously mentioned, this is due to the difficulties in handling and working with this material, as it does not precipitate homogeneously resulting in a non-compact pellet, but may also be due to the porous composition of this type of particle.

Considering similar depletion performances for MagP and Lx, both kinds of particle can be considered suitable for PCT immunodepletion in the immunoassay range (0–50 ng/mL). This is true in terms of amounts of immunoprecipitated PCT (up to 50 ng/mL) and of time to complete the pull-down process (2 h). In contrast, Lx appear to be less efficient at a higher concentration range (500 ng, Figure 2 and Appendix A).

The MagP furthermore offer easy and rapid handling using the magnetic force technology, while the Lx needs longer centrifugation steps.

Finally, the PGS seem to be less suitable for PCT immunodepletion purposes, due to a more difficult handling. Indeed, the amount of recovered PCT in some cases was higher than the input (Figure 4a), clearly indicating an artifact-affected quantitation.

### 2.4. rhPCT Is Sensitive to Oxidation

The structural prediction of PCT using Alphafold 2.3.2 (https://alphafold.ebi.ac.uk/entry/P01258, accessed on 15 April 2023) suggests a relatively simple structure of alternating helical and loop parts/intrinsic unstructured parts. In particular, these dynamic loops are highly accessible to post-translational modifications, proteolytic degradation, or denaturing processes. 

The oxidation can directly influence the immunoreactivity of the PCT antigen and can potentially impact the quantification by commercial immuno-assays. Data of the impact of oxidation on PCT immuno-quantification are supported by incubation of the rhPCT antigen with different concentrations of the reducing agent DTT (1–5 mmol, Figure 5 and Appendix A). Dependent on the reducing agent concentration, the loss of PCT immunogenicity can be partially prevented. With a DTT concentration of 2.5 mmol/L, approx. 50% of the oxidation-based loss of immuno-reactivity for pAbs and approx. 30% for mAbs can be recovered.

For this reason, the careful and fast handling of PCT immuno-enrichment and work-up, avoiding long incubation or agitation steps as well as long-term storage of samples, should be considered. By using nano-particles with a high surface–antibody ratio, an efficient depletion can be ensured (<2 h, see Figure 4). All three kinds of particles presented here show suitable depletion rates. However, PGS and Lx, probably due to intrinsic size heterogeneity and composition and associated retention of proteins, reveal less suitable characteristics in this regard.

It is also noteworthy that the PCT immuno-quantification based on pAbs (DiaSys PETIA) shows a lower sensitivity to oxidation (higher immuno-reactivity, significance criteria 5%) compared to the PCT immuno-quantification based on two mAbs (BRAHMS-Roche CLIA) (see light blue and dark blue bars in Figure 5). This observation confirms that PCT-specific pAbs are preferable analytical tools for immunoprecipitation, as they direct a wider range of variable epitopes, especially if oxidation or different post-translational modifications of an analyte need to be considered.

In summary, PCT seems to be sensitive to oxidation. This modification influences the immunogenicity of the antigen and may in turn lead to lower immuno-reactivity and an inaccurate quantification by commercial immuno-assays.

### 2.5. Precipitation Workflow in Different Human Materials and Evaluation of Unspecific Background

To get maximal sensitivity of an analytical method, the information about the ratio of the specific signal (antigen) to unspecific background is of high importance. Especially in the context of subsequent tryptic digest and mass spectrometric analysis, a high or varying amount of non-related protein impurities may impair the metrological quantification of the analyte.

Based on so far optimized components and depletion processes, precipitation efficiency and content of impurities of functionalized PGS, MagP, and Lx nano-particles were directly compared and characterized by SDS-PAGE and Coomassie stain. To avoid binding of endogenous, unrelated antibodies from the serum samples to the surface of nano-particles or to the specific anti-PCT antibodies, all particles were pre-incubated with human IgG Fc fragments (Appendix A) to saturate and block putative unspecific binding surfaces. As evident in Figure 6a,b, lane 4, MagP reveals the most efficient precipitation of rhPCT, spiked into human serum. The identity of rhPCT bands of the SDS-PAGE (Figure 6a) was additionally confirmed by MALDI-TOF analysis. Furthermore, MagP by far show the lowest unspecific protein impurities, followed by Sepharose and Lx particles (compare Figure 6, lane 4, MagP, with lane 5, PGS, and lane 6, Lx). The Lx seems to bind the greatest number of unspecific proteins. Probably, binding occurs due to the large surface and charge of the functionalized polystyrene particle matrix. This has already been reported in the literature on polystyrene particles used in immuno-agglutination [32,33] and seems to be applicable also to Lx nano-particles directly used for immuno-depletion attempts. In addition, PGS show significantly higher background compared to MagP depletion experiments, probably due to the porosity of the material and associated retention of unspecific proteins (Figure 6a, compare lane 5 to 4).

These observations were further confirmed by the use of HeLa cell extract in a comparable series of precipitation experiments (Figure 6b). The precipitation using HeLa cell extract indicates the possibility of this system in also quantifying PCT from human tissue lysate/cell extracts as well as the option of characterizing putative binding partners of PCT by subsequent proteomic analysis. Moreover, in this material the functionalized MagP by far show the most efficient precipitation of rhPCT and the lowest unspecific background. In contrast also in this setting, PGS clearly show elaborated unspecific protein content compared to MagP. Most unspecific proteins are precipitated by the Lx strategy (Figure 6b, compare lanes 4 to 5 and 6). In summary, functionalized MagP nanoparticles overall reveal the most efficient precipitation, combined with minimal unspecific background in human serum material but also in HeLa cell extract.

### 2.6. Relative Quantification of Immuno-Enriched PCT by LC-MS/MS

In a final step of evaluation, optimized functionalized nano-particles Lx, PGS, and MagP were characterized regarding the implementation of a tryptic “on-bead” digest of precipitated rhPCT, spiked into human serum and subsequent elution/removal of the resulting peptides and LC-MS/MS analysis.

A special focus was given to the assessment of the MS background of the three particles (Figure 7). As already observed by the SDS-PAGE gel analysis (Figure 6, compare lane 4 to 5 and 6), the MagP appear to offer the lowest profile of unspecific proteins also in LC-MS/MS analysis (Figure 7, black line). On the other hand, the Lx particles (red line) precipitate the highest amount of non-related compounds.

Furthermore, for the specific assessment and initial quantification of rhPCT in LC-MS/MS, six different peptides comprehensively covering the human PCT protein sequence (Appendix A) were used. All six obtained proteolytic peptides reveal approx. 5 to 10 times higher relative signal with MagP-derived LC-MS/MS quantification (Figure 8, black bars), compared to Lx (Figure 8, red bars) and PGS particles (Figure 8, blue bars). This clearly indicates that functionalized MagP nano-particles represent the most suitable quantitative immuno-enrichment system in terms of background and sensitivity of LC-MS/MS-based quantification.

## 3. Discussion

The comparable quantification of PCT in human sample material remains one of the most challenging issues in laboratory medicine. For this reason, in 2018 the International Federation of Clinical Chemistry (IFCC) initiated a dedicated working group to develop and validate a reference measurement procedure for absolute quantification of PCT by mass spectrometry (IFCC-WG PCT) [34].

Recently, Huynh et al. proposed a candidate mass spectrometry-based reference method for the quantification of PCT in human samples, based on the chemical precipitation of the antigen [25,26]. On the one hand, this method excludes the influence of the possible variable binding of the used antibodies onto the immuno-precipitation. On the other hand, however, it needs two sequential solid-phase extraction steps, which could affect the integrity of precipitation and also may interfere with the subsequent quantification process itself, as previously argued by Tölke et al. [27]. Indeed, only two tryptic peptides were precipitated by the method of Huynh et al., and only one of them could be used for metrological quantitation purposes, possibly not reflecting PCT variations (isoforms) or post-translational modifications.

In this work, we presented the development of an immunoprecipitation method based on the evaluation and the optimization of different kinds of immuno-functionalized nano-particles (Sepharose, magnetic, and polystyrene-based) utilizing a polyclonal antibody raised against the full-length human PCT antigen.

This work represents the upstream pre-analytical step for the mass spectrometry-based reference method and potentially the basis for further proteomic analyses to characterize PCT in detail.

The use of polyclonal antibodies to specifically immuno-precipitate PCT was chosen for several reasons. On the one hand, the first commercial PCT immunoassay (B·R·A·H·M·S PCT™ sensitive KRYPTOR™) employed sheep anti-PCT pAbs [21]. This immunoassay is still used arbitrarily as a relative reference test [24]. Second, PCT biochemistry is not yet well characterized. This is especially attributed to putative PCT interaction partners under physiological conditions or in the circumstances of bacterial-caused inflammation. An increased proteinase expression can be observed in almost every human tissue upon inflammation [35,36,37] and may regulate an inflammation-induced bioavailability and activity of interaction partners of PCT. So far, only a few interaction partners of PCT have been identified, such as the membrane-associated calcitonin gene-related peptide 1 (CGRP1) receptor and amylin 1 (AMY1) receptor [38] or the ankyrin repeat domain 11 protein (ANKRD11) [39]. Putative interaction partners of PCT could potentially influence the immunoassay reactivity by binding to PCT and blocking respective epitopes, contributing to the well-known discrepancies in immuno-quantification of PCT in clinical diagnostics [22,23,40]. Third, post-translational modification or denaturation of PCT that may occur during the onset and progression of sepsis may influence its immuno-reactivity.

In this regard, the present work demonstrated for the first time that the PCT molecule is prone to oxidation and its immuno-reactivity is significantly influenced by that. Indeed, upon treatments causing accelerated deterioration and oxidation (37 °C, 10 days), the antigen showed a decay in immuno-recovery of approx. 21% when analyzed via mAbs (BRAHMS-Roche CLIA assay), and ca. 15% decay with pAbs (DiaSys PETIA assay). This oxidation-derived effect could be partially reverted by incubation with increasing amounts of the reducing agent DTT. The incubation of rhPCT with 2.5 mmol/L of DTT resulted in the recovery of 91.5% for the pAbs-based test, and 85.4% for the mAbs-based one (Figure 5). These data confirm the assumption that pAbs are less affected compared to mAbs concerning rhPCT immuno-recognition (significance criteria 5%). This finding clearly indicates effects of oxidation on PCT, even though it remains to be proven how strong endogenous PCT is affected by this. However, as most immunoassay suppliers employ a full-length recombinant PCT antigen for preparing their calibrator solutions [24,41,42,43], the effect of oxidation on the recovery of recombinant PCT material is an important aspect to address.

The PCT structure, predicted by AlphaFold 2.3.2, illustrates the location of putative oxidized methionine residues (Figure 9, yellow highlighted residues) and of two cysteine residues, probably forming a disulfide bond (Figure 9, cyan-highlighted residues) [44,45].

The region of PCT putatively forming as cysteine bridge is strongly conserved among vertebrates, indicating important structural and functional aspects of PCT in this part [46]. The oxidation of cysteine residues and the associated structural dynamics in this area may lead to a varying accessibility of the respective epitopes. The final goal of the present work is to propose a reference method for the standardization of the multitude of different PCT immunoassays on the market. Consequently, the use of pAbs to selectively precipitate PCT from a sample could overcome the mentioned hurdles, guaranteeing a robust multiplicity of binding and immuno-enrichment, independently of the antigen conditions.

In addition, differences in immunoreactivity between recombinant and wild-type PCT have recently been reported for immunoassays based on mAbs in comparison to pAb-based ones [24], strengthening the differences between these two antibody-based approaches.

Besides the sound selection and characterization of antibodies, a special focus of this work was given to the carrier system to immobilize the antibodies. For fast and efficient precipitation of PCT for subsequent metrological analysis, the chosen system needs to combine a large immuno-functionalized surface with robust and reproducible handling, which needs to be realized with usual laboratory instruments. To this end, functionalized microspheres ranging from 0.35 to approx. 90 µm with quite some diverging capacity in binding immunoglobulins (up to 1800 µg, PGS) were used for the evaluation (Table 1 and Figure 1). These functionalized nano-particles enable direct binding of the antibody by chemical cross-linking (Lx) or indirectly binding the antibodies via protein G/A linker proteins (PGS or MagP).

As this interaction takes place through the Fc part of the antibody, it leads to a better steric orientation of the bipartite paratopes and a higher precipitation rate of the antigen. Covalent coupling of the antibodies and the Protein G linker of the nanoparticles might lower the antibody binding affinity for the antigen by partially denaturing the antibody´s structure during the cross-linking process. For this, a direct comparison of the precipitation efficiency of a covalent vs. non-covalent cross-link was evaluated. According to our data, both conditions appeared to precipitate similar amounts of PCT antigen (Figure 2a,b for the PGS, Figure 2c,d for the MagP).

On the other hand, the binding of the antibodies to the negatively charged Lx particles randomly involves one of the two amino-terminal CDRs, as previously described by Puertas et al. [31]. This in turn results in a lower immunoprecipitation efficiency by blocking the respective paratope. Nonetheless, the smaller particle diameter and the associated large surface area of the Lx (approx. 0.35 µm) compared to that of MagP (approx. 2.8 µm) and PGS (approx. 90 µm) compensate for the lower immunoprecipitation grade.

Indeed, quantitation of the immuno-depletion efficiency of the three particles by rhPCT spiked in serum in the immunoassay analytical range (0.5–50 ng/mL) unequivocally demonstrated that the Lx, as well as the MagP particles, did precipitate nearly 100% of the input rhPCT within two hours of incubation (Figure 4b,c). In contrast, the PGS showed implausible depletion kinetic (Figure 4a, varying PCT quantification due to the difficult handling of the respective particles), probably due to the less efficient depletion or incomplete centrifugation or unintentional resuspension of the PSG particles upon removal of the supernatant after centrifugation. Indeed, Appendix A visually confirms a less defined and compact pellet on the bottom of the tube formed by the PGS (lane 1) in comparison to MagP and also Lx (lanes 2 and 3).

In light of the observed oxidation of PCT and its concomitant decay in immuno-reactivity, associated with this post-translational modification, it is necessary to consider the impact of the upfront handling time of PCT. Long-time storage of sample material as well as complex and long-lasting processes in pre-analytical steps of LC-MS/MS quantification should be reduced. To this end, 2 h of overall handling time as well as the loss-free and robust performance of immuno-functionalized MagP is beneficial.

Comparable data on the precipitating efficiency were obtained by rhPCT quantitation via iBright (see Appendix A). Even though the amount of each immuno-functionalized particle was normalized to precipitate 1000 ng of antigen (Figure 2), the PGS only recovered approx. 50% of the rhPCT input. Similarly, the Lx particles recovered approx. 80% of the rhPCT input. In both cases, the duplicates of the depletion capacity displayed a significant difference (Δ = 35 ng, PGS and Δ = 25.5 ng, Lx). In clear contrast, the MagP showed robust performances, both in terms of recovery (552.8 ng vs. 500 ng input) and repeatability (Δ = 5.5 ng). The recovered amount, slightly higher than the input, could be ascribable to imprecision in handling, which however does not affect the overall conclusion about the better performances of this kind of particle over PGS and Lx.

The observed results are in good agreement with previous reports proposing the use of magnetic particles to enrich samples as a preparation step for subsequent mass spectrometric analysis [47,48]. In particular, Whiteaker et al. [49] applied this methodology to α1-antichymotrypsin and TNF-α, the latter having very low blood concentrations similar to those of PCT.

Especially concerning low abundant analytes, putative sources of background that may lower the specific signal of later quantification and thus the method sensitivity need to be characterized. In the present work, unspecific interactions from human samples (denatured proteins, unspecific binding proteins, lipids, etc.) were preventively quenched upon specific blocking of the immuno-functionalized nanoparticles. For example, human Fc fragments were used to specifically quench free Protein G, to avoid binding of rheumatoid factors via Fc–Fc interactions of unrelated antibodies derived from the human serum sample (Appendix A).

Even though the LC-MS/MS quantification used here is very specific *per se*, by using an isotope labeled spiked internal standard protein the most problematic aspects in relation to the unspecific reactions lay in the pre-analytical steps of sample preparation and immunoprecipitation. Indeed, various putative interfering components may mask or block antibody-binding sites and for this reason may reduce the efficiency of immunoprecipitation. For this, intensive work was undertaken on the evaluation and characterization of handling conditions (amounts of beads to be used, antibodies, particles, etc., see Figure 2) and the use of blocking components for the quantitative immunodepletion of PCT (see Figure 6 and Appendix A).

The background of unspecific proteins as well as a specific amount of immuno-precipitated rhPCT were also quantified by proteolytic on-bead digestion and subsequent LC-MS/MS analysis (Figure 7 and Figure 8). The ratio of unspecific background and specific PCT quantification revealed striking differences (Figure 6 and Figure 7). PGS and especially Lx (Figure 7, red profile) showed higher unspecific background and revealed low specific signal in LC-MS/MS quantification. In contrast, MagP along with low unspecific background achieved up to three magnitudes higher specific signal, compared to Lx or PGS.

The LC-MS/MS results on unspecific and specific binding were confirmed by the direct comparison of SDS-PAGE analysis for all three nanoparticles (Figure 6). Indeed, while MagP presented good recovery of rhPCT, PGS and especially Lx showed lower antigen recovery as well as an increased unspecific background from human serum and HeLa cell extract. Especially, the background observed for the Lx confirmed previous reports about polystyrene particles, which are prone to unspecific interactions with serum proteins [33,50,51,52]. Additionally, the possible variation of patient-derived samples, especially in the highly deregulated setting of an inflammation, can result in problematic sample composition (dysregulation of antibody content [53], strongly varying overall concentration of proteins [54,55,56], lipids as well as a high degree of newly expressed inflammation proteins or denatured proteins due to fever [57]). To assess this degree of variation in a representative cohort of samples will be a necessary next step in the evaluation of the LC-MS/MS reference methods, in comparison to PCT immunoassays, as already highlighted and planned by the IFCC working group on Standardization of Procalcitonin Assays (WG-PCT) [34].

While the precise knowledge of PCT concentrations in human specimens does essentially contribute to the accurate sepsis diagnosis, classical proteomic work is frequently focused on the specific antigen immuno-precipitation from a variety of cell lines and their extracts. The major aspect of the cell extract-derived analysis is to identify and characterize existing or new interaction partners within a protein network. The characterization of the HeLa-cell extract-derived recovery of rhPCT presented here (Figure 6b) in this context will help to characterize modifications of endogenous PCT and its interaction partners in the future.

## 4. Materials and Methods

### 4.1. Primary Antibodies, Cell Line, Serum Sample Pools

Anti-PCT polyclonal antibodies, raised in goat and immunopurified by full-length recombinant PCT antigen, were provided by DiaSys Diagnostic Systems GmbH (Holzheim, Germany). HeLa cervical carcinoma cells for the analysis of protein interaction were purchased from IPRACELL (B-7000 Mons, Belgium). Serum samples and sample pools were provided by DiaServe Laboratories GmbH (Iffeldorf, Germany). Sample pools were frozen and stored at −80 °C until respective measurements.

### 4.2. Covalent Coupling of Antibodies to Nano-Particles

Antibodies were bound to protein G Sepharose particles (Protein G Sepharose 4 Fast Flow, recombinant protein G lacking the albumin-binding region, covalently cross-linked to agarose, Cytiva/GE Healthcare Life Sciences/Danaher Corporation, Washington, DC, USA) (PGS) and protein G magnetic particles (Dynabeads^TM^ Protein G for immunoprecipitation, superparamagnetic core particles with covalently linked recombinant protein G, Invitrogen—Thermo Fisher Scientific Inc., Waltham, MA, USA) (MagP) by incubating PCT-specific polyclonal antibodies together with the respective nano-particles in 1 mL PBS buffer, containing Triton X-100 (0.01%, *w*/*v*) at 4 °C for 8 h on a roller mixer (40 rpm).

For the subsequent cross-linking, PGS and MagP were equilibrated twice with 0.2 M Na-tetraborate, pH 9.0, and cross-linked with 0.052 g dimethylpimelimidate (DMP) in 10 mL borate buffered for 60 min at room temperature. The cross-linking process was stopped with 100 mmol/L Tris/HCl, 150 mmol/L NaCl, 0.01% (*w*/*v*) Triton X-100, pH 7.5. Afterward, particles were incubated with 50 µg human IgG Fc-fragments (Purified Human Immunoglobulin G (IgG), Fc-Fragment, Meridian Life Science, Inc., Memphis, TN, USA) in 100 mmol/L Tris/HCl, 150 mmol/L NaCl, 0.01% (*w*/*v*) Triton X-100, pH 7.5, for 8 h at 4 °C on a roller mixer to block unspecific binding of unrelated proteins with Protein G. After incubation, particles were washed twice with 100 mmol/L Tris/HCl, 150 mmol/L NaCl, 0.01% (*w*/*v*) Triton X-100, pH 7.5 buffer. Ready-to-use, functionalized anti-PCT polystyrene-based particles (Latex, Lx) were provided by DiaSys Diagnostic Systems GmbH (Holzheim, Germany).

### 4.3. Analysis of Particles, Size Determination, Calculation of Binding Capacities

Particles (PGS, MagP, Lx), after covalent coupling to anti-PCT, were resuspended in water and sonicated. Approximately 20 µL of each suspension was applied on a glass slide and capped by a cover slip. A Leica DM1000 optical microscope (Biosystems Switzerland AG, Muttenz, Switzerland), phase contrast mode, using 100× (PGS), 400× (MagP), and 1000× (Lx) enlargement, was used in combination with a Leica DFC420 digital camera (Twain Version 7.1.0.0, Fx Lib 5.0.4.36, Muttenz, Switzerland) for documentation. Size characterization of the particles was performed by using the analytical microscope software Leica Application Suite (Version 3.5.0, Muttenz, Switzerland). To assess surface integrity or porosity, particles were stained by Coomassie brilliant blue G-250 solution and visualized by phase contrast microscopy as described above.

To assess the depletion capacity in the analytical range of most commercial PCT immunoassays (0 to approx. 50 ng/mL), PBS buffer containing 3 mg/mL BSA was spiked with 50 ng/mL, 5 ng/mL, and 0.5 ng/mL rhPCT to reach a total reaction volume of 1 mL. Highly purified untagged full-length recombinant human PCT (rhPCT, refer to Appendix A for the detailed sequence of human PCT) was provided by HyTest Ltd. (Turku, Finland). Subsequently, the nano-particles were added as follows: PGS 15 µL; MagP 50 µL; Lx 150 µL (concentration 1% *w*/*v*). The incubation was conducted for 6 h. Every 2 h, a sample was taken and the particles separated as described above. The supernatant was analyzed by a clinical chemistry analyzer as described later on to quantify rhPCT, not yet bound by the particles. The recoveries of rhPCT in the supernatant in ng/mL were plotted in a graph vs. the time of incubation using Microsoft Excel 2016 (v. 16.0.4639.1000) 64-Bit (Microsoft, Redmond, WA, USA).

### 4.4. Preparation of Cell Extract, Serum, and Immuno-Enrichment

Total cell extracts of HeLa cells were generated by sonication of cell pellet resuspended in lysis buffer 50 mmol/L Tris/HCl, 200 mmol/L NaCl, 0.25% (*w*/*v*) Triton X-100, 5 mmol/L EDTA, 0.5 mmol/L dithiothreitol (DTT), containing one cOmplete ULTRA Tablet, Mini, EASYpack protease inhibitor, pH 7.5 (Roche Diagnostics GmbH, Mannheim, Germany). After sonication, crude cell lysate was centrifuged at 20,000× *g* for 20 min at 4 °C. Clarified supernatants were further sequential filtered by low protein binding filters (Filtropur S 0.45 µm and S 0.2 µm, SARSTEDT AG & Co. KG, Nümbrecht, Germany). If serum was used, the clearing was performed by 10 min centrifugation at 7.500 rpm and subsequent filtration (Filtropur S 0.45 µm and S 0.2 µm, SARSTEDT AG & Co. KG, Nümbrecht, Germany). Equal amounts of HeLa cell extract or serum, respectively, were spiked with rhPCT. MagP, PGS, and functionalized Lx particles were incubated for 2 h with human serum or HeLa extract at 4 °C on a tube roller. Amounts of all used immuno-functionalized particles were upfront normalized to bind identical quantities of rhPCT. After incubation, particles were washed 3 times with 50 mmol/L Tris/HCl, 200 mmol/L NaCl, 5 mmol/L EDTA, 0.25% (*w*/*v*) Triton-X 100, pH 7.5. After each washing step, the beads were separated by centrifugation for 2 min at 1500 rpm (PGS), 5 min at 13,500 rpm (Lx), or by magnetic forces using a neodymium magnet to fix MagP to the wall of the reaction tube (approx. 30 s). The bound total protein content of the different matrices was eluted from the respective beads boiling the samples in 2× SDS sample buffer. The total content of immuno-enriched protein was resolved by 15% SDS-PAGE gels. The input of serum and cell extract before immuno-depletion, after depletion, respective aliquots of washing steps, and eluted total content of precipitated protein were visualized by Coomassie stain of the SDS-PAGE.

### 4.5. Relative Quantification of Immuno-Enriched PCT by MS

The mass spectrometric analysis was performed on an LC-MS/MS system comprising an UHPLC system (Agilent 1200) and an electrospray ionization triple quadrupole mass spectrometer (5500 QTRAP from Sciex, Framingham, MA, USA), operated in positive ion mode. The Acquity HSS T3 (1.8 µm; 2.1 × 50 mm) column was used (Waters corporation, Milford, CT, USA). Eluent A was H_2_O with 0.1% formic acid and eluent B was CH_3_CN containing 0.1% formic acid. The injection volume was 20 µL using a 100 µL sample loop. The peptide separation took place at room temperature with a 15 min gradient and a flow rate of 300 µL/min. Briefly, 5% eluent B was held for 2 min. Subsequently, stepwise eluent B gradients were applied as follows: from 2 to 7 min, 5 to 20%; from 7 to 8 min, 20 to 30%; from 8 to 12 min, 30 to 45%; from 12 to 12.1 min, 45% to 95%. A total of 95% eluent B was held up to 14 min and again decreased to 5% between 14.1 and 15 min. The ionization took place by ESI in positive mode with a Turbospray source (Sciex, Framingham, MA, USA) and at a temperature of 600 °C. Further settings as follows: curtain gas: 40, source gas 1:80, source gas 2:80, spray voltage: 5000 V, entrance potential of 8 V. The resolution was set for both Q1 and Q3 to unit.

Following, we report the detailed upstream optimization of the immuno-precipitation process. Quantification of recombinant human PCT, immuno-enriched by the three particles, PGS, MagP, and Lx, was performed by tryptic “on-bead” digestion. The matrices were washed and equilibrated in 50 mmol/L Tris/HCl buffer, pH 8.0. For tryptic digest, 60 µL buffer (50 mmol/L Tris/HCl, pH 8.0), containing 8 µg trypsin from bovine pancreas, N-tosyl-L-phenylalanine chloromethyl ketone (TPCK) treated to inactivate extraneous chymotryptic activity (Sigma-Aldrich, cat. T1426, St. Louis, MO, USA) was added to the respective precipitated nano-particle and incubated at 37 °C for 2 h in a PCR-thermoblock. The digest was centrifuged by a spin filter (Costar^®^ Spin—X Centrifuge Tube filter; 0.22 µm cellulose acetate in 2.0 mL polypropylene tube, RNase/DNase free, Corning Incorporated Salt Lake City, UT, USA) for 5 min, 10,000× *g*, RT. A total of 60 µL of this filtrate was added with 14 µL of 1% formic acid (Supelco—Sigma-Aldrich, St. Louis, MO, USA) in 25% acetonitrile (Supelco—Sigma-Aldrich, St. Louis, MO, USA) and used for the subsequent LC-MS/MS instrument (LC module, Model 1200, Agilent Technologies, Santa Clara, CA, USA; MS module, Triple quadrupole mass spectrometer 5500 QTRAP, AB Sciex LLC, Framingham, MA, USA) for further analysis. The qualitative identification of the immuno-precipitated rhPCT was performed by cutting out the respective band of approx. 13 kDa from the 15% SDS-PAGE gel and identifying the protein after tryptic digest (70 µL Trypsin, for 30 min in ice, and subsequent resuspension in ammonium bicarbonate 25 mmol/L) by MALDI-TOF analysis, using a Bruker autoflex^TM^ speed MALDI-TOF, with 1 kHz Smartbeam Laser (Bruker Corporation, Billerica, MA, USA).

### 4.6. Assessment of Immuno-Reactivity of Soluble PCT and Quantification of Stained PCT Resolved by SDS-PAGE

Quantification and kinetics of depletion of rhPCT from human serum pools or of soluble rhPCT were conducted using the PCT FS quantification kit of DiaSys Diagnostic Systems GmbH (Holzheim, Germany) on a cobas c 501 clinical chemistry analyzer from Roche Diagnostics GmbH (Mannheim, Germany). All reagents, calibrators, and quality controls on the PCT FS quantification kit were provided by DiaSys (Holzheim, Germany). The PCT FS PETIA methods required a 6-point calibration with concentrations between 0 and approx. 50 µg/L (0, 0.84, 3.49, 10.93, 23.56, 53.94 µg/L). PETIA PCT FS method has two levels of internal quality controls: level 1, 0.93 µg/L, and level 2, 10.9 µg/L.

The concentrations of the PCT solutions (50 ng/mL, 5 ng/mL, and 0.5 ng/mL) used for the pull-down experiment were initially quantified with the PETIA PCT FS assay and adjusted if necessary. At each incubation time, the PCT amount not bound by the beads was quantified with the same test. The pull-down efficiency for each type of particle at each measuring point was calculated in percentage recovery in reference to the initial amount of PCT according to the following calculation:$$Binding\;capacity\left(\%\right)=100-\frac{{Output\;amount}_{\left(at\;time\;x\right)}\times 100}{{Input\;amount}_{\left(at\;time\;0\right)}}$$

Further quantification of immunoprecipitated PCT was performed by denaturing SDS-PAGE and subsequent Coomassie staining in reference to a defined dilution series of free rhPCT on the same gel. Assessment and relative quantification of protein contents were performed by high-content imaging and analysis iBright CL1000 System (Thermo Fisher Scientific Inc., Waltham, MA, USA), combined with the respective analytical iBright software (Version 4.0.1, Thermo Fisher Scientific Inc., Waltham, MA, USA).

Similarly to the concentrations in the relevant medical range of PCT, the depletion efficiency percentage of the three types of particles in a PCT concentration range up to 500 ng was calculated according to the formula reported below. Differences bigger than 5% were considered significant.
$$Depletion\;efficiency\left(\%\right)=\frac{O{utput\;amount}_{\left(from\;iBright\right)}}{Input\;amount}\times 100$$

### 4.7. Assessment of the Impact of Oxidation on the Immunogenicity of rhPCT

The effect of partial oxidation of rhPCT, potentially impacting its recovery upon IP, was demonstrated by incubation of rhPCT at elaborated temperature (37 °C), with and without increasing amounts of the reducing agent dithiothreitol (DTT). Six solutions containing increasing amounts of rhPCT (0, 0.944, 4.8, 12.2, 24.6, 54.7 ng/mL) were spiked with an increasing amount of DTT (0, 1, 2.5, 5 mmol/L). One aliquot of each rhPCT solution was incubated for 6 or 10 days at 37 °C, respectively. To assess the impact of treatment on the subsequent immuno-quantification, PCT FS (DiaSys Diagnostic GmbH, Holzheim, Germany) and Elecsys^®^ BRAHMS Procalcitonin (PCT) (Roche Diagnostics, Mannheim, Germany) were evaluated on a BioMajesty™ JCA-BM6010/C (PCT FS test) or a cobas e 411 (BRAHMS-Roche PCT assay). The impact of oxidation on the immunogenicity of the PCT antigen was evaluated as recovery of PCT by the two immunotests calculated as percentage in reference to the untreated PCT solution. The average of the recovery of all PCT solutions was calculated according to the formula reported below. Differences bigger than 5% were considered significant.
$$Oxidation\;rate\left(\%\right)=100-\left\{\frac{\sum_{N(=5\;PCT\;conc.)}\left[\left(\frac{{PCT\;recovery}_{day\;10}}{{PCT\;recovery}_{day\;0}}\right)\times 100\right]}{N(=5\;PCT\;conc.)}\right\}$$

Furthermore, the recovery of immunogenicity upon the use of the anti-oxidative agent (DTT) was calculated as follows:$$\begin{array}{l} Recovery\;of\;loss\;of\;immunogenicity\left(\%\right)\\ =\left(\frac{Loss\;of\;immunogenicity\;at\;day\;10\;without\;DTT\left(\%\right)\;-\;Loss\;of\;immunogenicity\;at\;day\;10\;with\;DTT\left(\%\right)}{Loss\;of\;immunogenicity\;at\;day\;10\;without\;DTT\left(\%\right)}\right)\times 100 \end{array}$$

### 4.8. Structural Prediction by AlphaFold 2.3.2

Structural predictions were performed with the program “AlphaFold 2.3.2” (DeepMind, Google LLC, Mountain View, CA, USA), based on the human PCT NCBI reference sequence NP_001365878.1 (calcitonin isoform CT preproprotein), excluding the 25 amino acid signal peptide [58]. The representative structure shown here is the one with the highest confidence of the five prediction versions calculated. Visualization was performed by the software PyMOL 2.5 (https://pymol.org/, accessed on 15 April 2023).

## 5. Conclusions

The present work showed for the first time the in-parallel optimization and characterization of three immunoprecipitation methods for the enrichment of PCT-containing samples (Table 2). It could be demonstrated that the magnetic particles, functionalized by polyclonal PCT-specific antibodies, offer the best performances in terms of specific and reproducible quantitative binding of PCT in human serum or HeLa cell extract. The method proposed is robust, reliable, and suitable for the immuno-enrichment of PCT for subsequent quantitative mass spectrometry or proteomic analyses.

## 6. Patents

TM and MG are named as inventors in an international patent application (PCT/EP2021/051577), claiming the manufacturing and use of the described PETIA for the quantification of PCT.

## Figures and Tables

**Figure 1 ijms-24-10963-f001:**
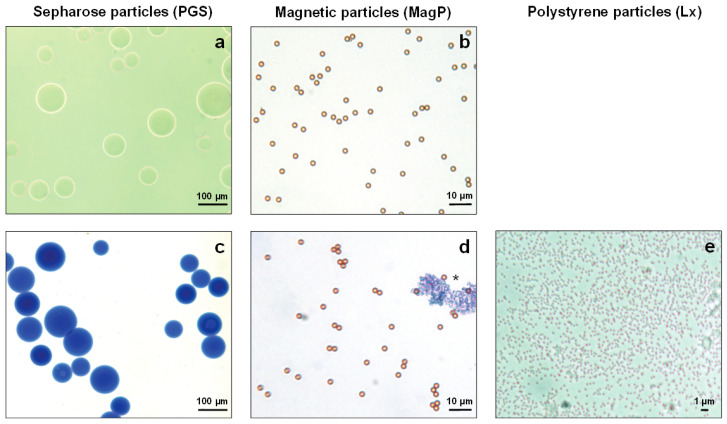
(**a**–**e**) Particles were analyzed by phase contrast microscopy (**a**,**b**) to evaluate size distribution and homogeneity of particle size. Latex particles were analyzed by light scattering as they were indistinguishable at the optical microscope without staining. To assess surface integrity and porosity, particles were stained by Coomassie brilliant blue G-250 solution and visualized by phase contrast microscopy (**c**–**e**). Leica DM1000 light microscope phase contrast (lens 100×, 400×, 1000×), digital camera Leica DFC420, and analytical microscope software Leica Application Suite. The asterisk indicates an aggregate of the Coomassie brilliant blue G-250 dye.

**Figure 2 ijms-24-10963-f002:**
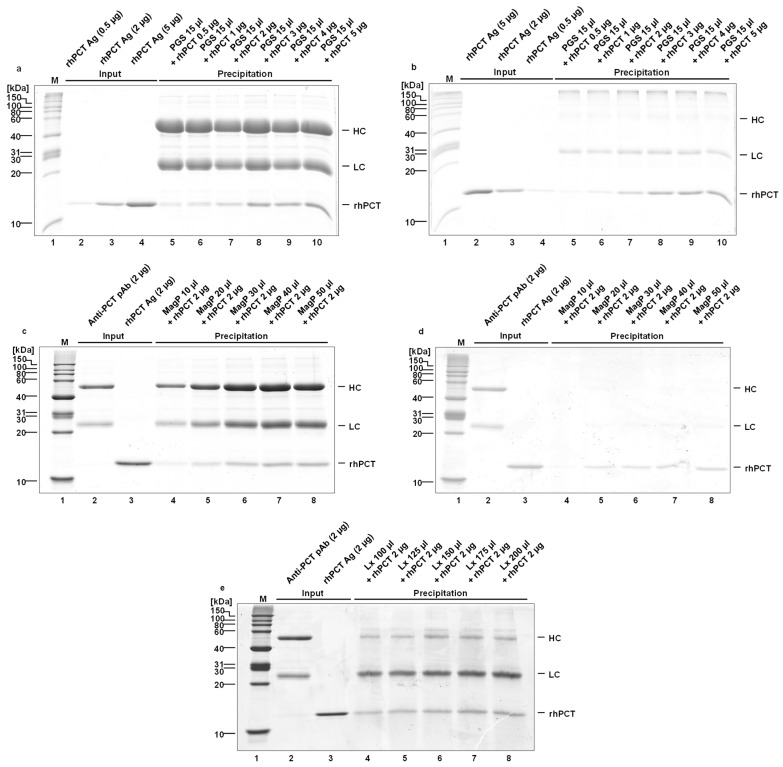
(**a**–**e**) Optimization of the depletion capacity of the three nano-particles (PGS, MagP, and Lx) and quantification by SDS-PAGE. Protein molecular weight marker (M). rhPCT Ag and Anti-PCT Ag indicate the respectively used quantity of inputs. The quantity of the input rhPCT band was used to quantify the immuno-precipitated rhPCT by the different nano-particles. (**a**) PGS, non-covalently bound anti-PCT antibodies. (**b**) PGS, covalently bound anti-PCT antibodies. (**c**) MagP, non-covalently bound anti-PCT antibodies. (**d**) MagP, covalently bound anti-PCT antibodies. (**e**) Titration of Latex particles, immuno-functionalized with anti-PCT for optimization of precipitation efficiency.

**Figure 3 ijms-24-10963-f003:**
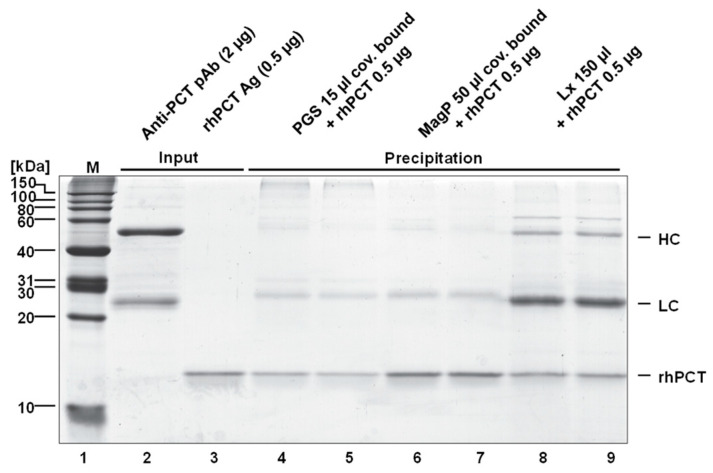
Duplicate recovery of all three particles. The input of rhPCT (500 ng, lane 3) has been used as reference to calculate the relative recovery of rhPCT precipitation, by quantifying band intensity via iBright software (see Appendix A). Lane 1 shows protein marker, lane 2 input of used PCT-specific antibody. Lanes 4, 5 show PGS, lanes 6, 7 show MagP, and lanes 8, 9 show Lx duplicate precipitations.

**Figure 4 ijms-24-10963-f004:**
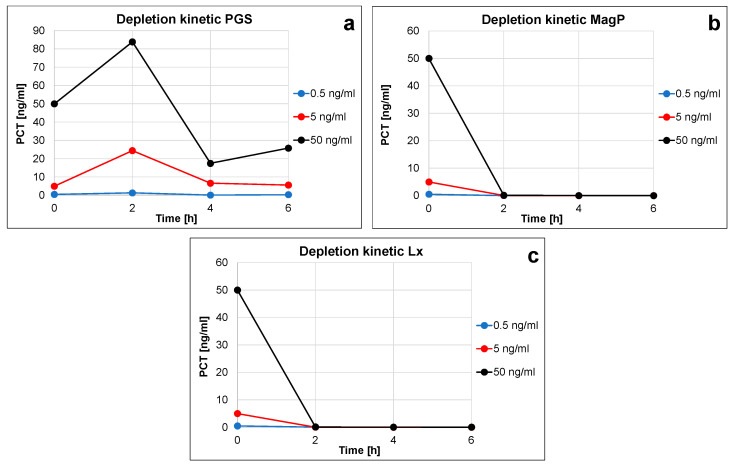
(**a**–**c**) PCT depletion kinetics of the three particles ((**a**). PGS, (**b**). MagP, (**c**). Lx). The rhPCT remaining in the supernatant after immuno-depletion was quantified by the PCT FS assay. Depletion efficiency was assessed at 2, 4, and 6 h upon incubation with respective immuno-functionalized nano-particles.

**Figure 5 ijms-24-10963-f005:**
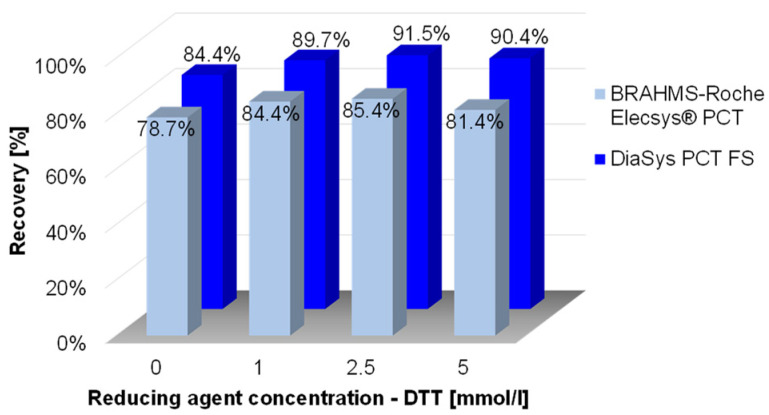
Recovery of rhPCT after 10 days of incubation at 37 °C with increasing amounts of reducing agents (DTT). The recovery was analyzed using two different immunoassays: the CLIA PCT by Roche, based on two mAbs, and the PETIA by DiaSys, based on pAbs. The PETIA seems less sensitive to oxidation. DTT concentration 2.5 mmol/L mediates the lowest reduction of immuno-reactivity (8.5% DiaSys PETIA and 14.6% Roche CLIA).

**Figure 6 ijms-24-10963-f006:**
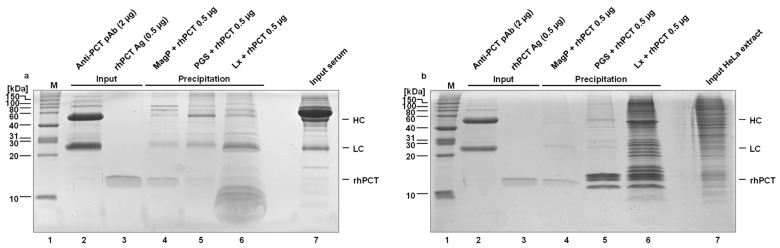
(**a**,**b**) Comparison of the precipitation efficiency and ratio to unspecific background by different immuno-functionalized microsphere matrices towards (**a**) Human serum and (**b**) HeLa cell extract.

**Figure 7 ijms-24-10963-f007:**
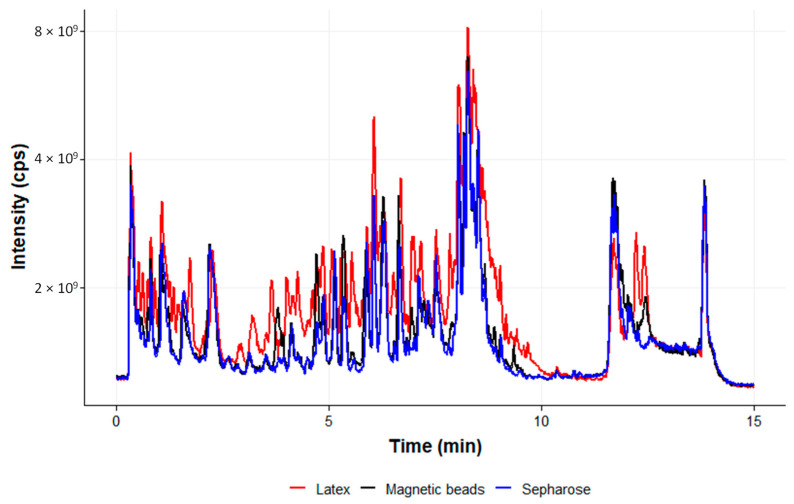
Comparison of background in mass spectrometry analysis by the three particles when human serum is used for spiking rhPCT. Total ion chromatogram scan *m*/*z* 200–*m*/*z* 1200. Black: profile of MagP; red: profile of Lx; blue: profile of PGS particles.

**Figure 8 ijms-24-10963-f008:**
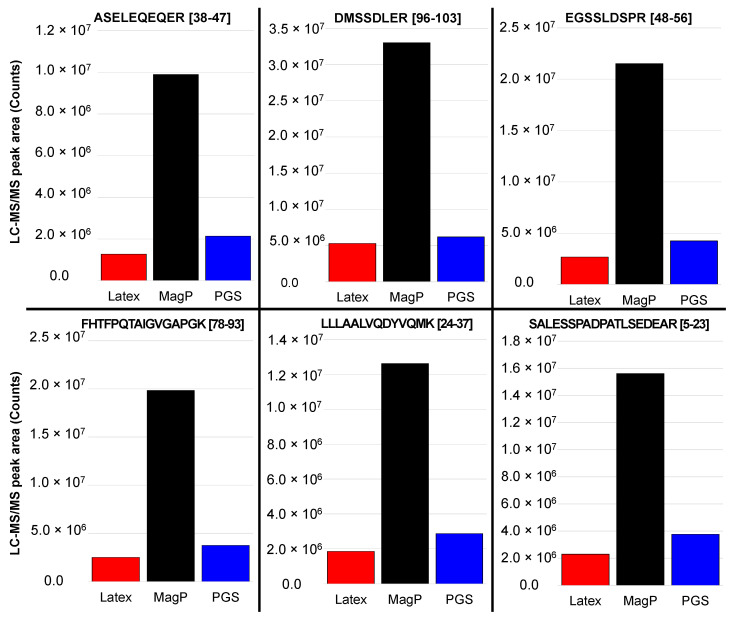
Peak areas for the six main peptides generated by the tryptic digestion of rhPCT followed by LC-MS/MS analysis. The three kinds of particles (red: Lx; black: MagP; blue: PGS) were used for the immuno-precipitation of rhPCT. Tryptic digestion was performed directly on the respective nano-particles.

**Figure 9 ijms-24-10963-f009:**
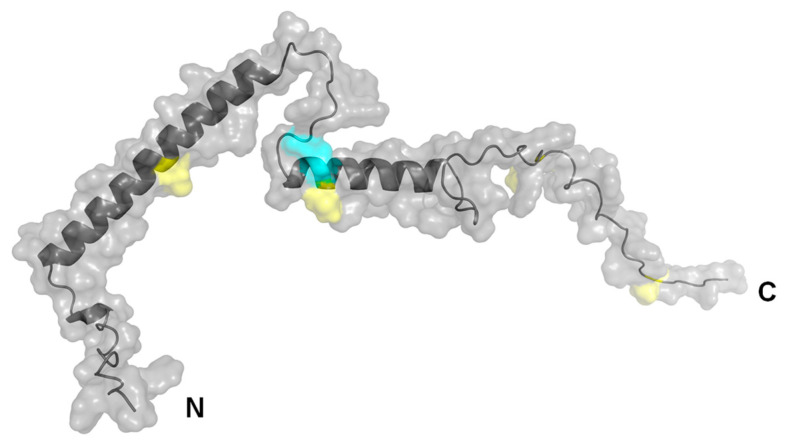
Structure of human PCT predicted by AlphaFold 2.3.2, lacking the 25 amino acids N-terminal signal peptide. Cysteine residues marked in cyan possibly form a disulfide bridge. The four methionine residues, also offering a potential oxidation site, are marked in yellow.

**Table 1 ijms-24-10963-t001:** Overview of used microspheres and properties of binding for the three nano-particles characterized in the present work.

	Sepharose Particles (PGS)	Magnetic Particles (MagP)	Polystyrene Particles (Lx)
Diameter	~90 µm	2.8 µm	0.350 µm
Structure	porous/open	compact/close	compact/close
Representation of coupled antibodies	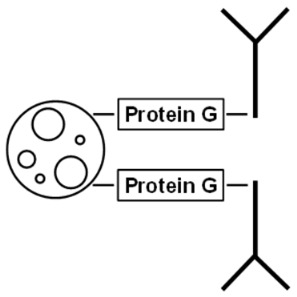	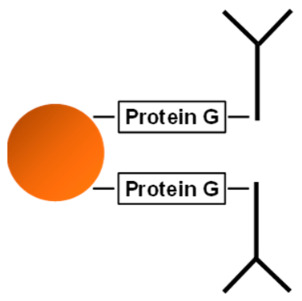	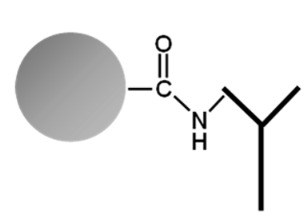
Ig binding capacity	100 µL PGS bind approx. 1800 µg rabbit IgG	100 µL MagP isolate approx. 25–30 µg human IgG	n/a

**Table 2 ijms-24-10963-t002:** Summary of the immunoprecipitation characteristics of the three particles described in this work. The MagP by far showed the best performances in terms of specific PCT immunoprecipitation and the lowest unspecific binding (green = good/best performances; yellow = fair/sufficient performances; red = poor/unsatisfactory performances).

	PGS	MagP	Latex
Antibody binding capacity	Good	Fair	Good
PCT depletion efficiency	Good	Good	Good
Easy handling/washing	Fair	Good	Fair
Reproducibility	Poor	Good	Fair
Unspecific interactions with human serum (SDS-PAGE)	Fair	Good	Poor
PCT depletion efficiency in human serum (SDS-PAGE)	Poor	Good	Poor
Unspecific interactions with human serum (mass spectrometry)	Good	Good	Fair
PCT depletion efficiency in human serum (mass spectrometry)	Poor	Good	Poor
Unspecific interactions with HeLa cell extract (SDS-PAGE)	Fair	Good	Poor
PCT depletion efficiency in HeLa cell extract (SDS-PAGE)	Poor	Good	Poor

## Data Availability

All the data are available upon approval of the corresponding author.

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
