# Peer review of "Comprehensive Comparison of the Capacity of Functionalized Sepharose, Magnetic Core, and Polystyrene Nanoparticles to Immuno-Precipitate Procalcitonin from Human Material for the Subsequent Quantification by LC-MS/MS"

_ijms, 2023, doi:10.3390/ijms241310963_

Round 1

Reviewer 1 Report

The authors described the characterization of three immunoprecipitation methods for the enrichment of protein procalcitonin-containing samples. They have compared the methods and optimized the steps of the procedure.

The paper is quite important, especially for the possible applications in proteomics.

The authors have published other articles in this field in which are competent and expert.

The experiments appear correctly conducted, the statistics is not clearly stated, and therefore I suggest introducing in the experimental section the statistical methods used.

The reference 23 and 35 are the same

Reviewer 2 Report

The authors present a very interesting research study with encouraging results. I recommend the following comments be addressed by the authors before the manuscript can be considered for publication.

1) In section "2.1. Covalent coupling of antibodies to nano-particles 101 The three nano-particles characterized in this work (PGS, MagP, and Lx) employ different binding schemes to capture the antibodies. In Error! Reference source not found., 103 the binding, as well as the main characteristics of the three different nano-particles are 104 summarized" please add the appropriate reference. Sections 2.2 and 2.3 also have references missing. Please revise according to the journal guidelines.

2) In figure 7 y axis needs units for the area

3) Please include scale bars in images shown in Figure 1

4) In figure 3 it is unclear how the authors conclude that "In summary, the overall depletion efficiency of the three immunofunctionalized 249 nano-particles PGS, Lx, MagP, clearly revealed the best performances for MagP, combined 250 with easy handling using magnetic force technology and also fast and quantitative depletion on human serum samples within 2 h of incubation." please include a discussion the plots for MagP and Lx look identical is this an error?

5) In the figure 4 data how exactly are 30% of the mAbs recovered? There should be a short explanation.

6) What degree of non-specific binding distortions could impact the data? What was done to ensure binding specificity?

Only minor proof reading is necessary
